# Generalized Real-space Chern Number Formula and Entanglement Hamiltonian

Ruihua Fan[†1], Pengfei Zhang[†2,3], and Yingfei Gu[*4,3]

[1]*Department of Physics, Harvard University, Cambridge, MA 02138, U.S.A.*
[2]*Department of Physics, Fudan University, Shanghai, 200438, P.R.C.*
[3]*California Institute of Technology, Pasadena, CA 91125, U.S.A.*
[4]*Institute for Advanced Study, Tsinghua University, Beijing, 100084, P.R.C.*

May 9, 2023

## Abstract

We generalize a real-space Chern number formula for gapped free fermions to higher orders. Using the generalized formula, we prove recent proposals for extracting thermal and electric Hall conductance from the ground state via the entanglement Hamiltonian in the special case of non-interacting fermions, providing a concrete example of the connection between entanglement and topology in quantum phases of matter.

# Contents

---

[†]They contribute equally to this work.
[*]guyingfei@gmail.com

# 1 Introduction

Two-dimensional gapped systems can exhibit rich topological phenomena. For example, the quasi-particle excitations in topologically ordered states possess anyonic statistics [1]; the chiral edge modes of gapped ground states with broken time-reversal symmetry lead to quantized transport coefficients such as the electric and thermal Hall conductance [2,3]. As entanglement is playing an increasingly important role in unifying different branches of quantum physics, it is of foundational interest to explicitly relate these topological properties to the complex pattern of entanglement in the ground state.

A classic example along this line is the topological entanglement entropy [4–6], which is the constant piece in the von Neumann entropy $S(\rho_A)$ of the reduced density matrix $\rho_A$ on a subregion $A$ of a gapped ground state. The topological entanglement entropy probes the total quantum dimension, a topological invariant that reflects the total "size" of the superselection sectors. Later, a more refined characterization was proposed by Li and Haldane [7]. The basic idea was to regard the spectrum of the *entanglement Hamiltonian*

$$K_A := -\ln \rho_A \qquad (1.1)$$

as a resolution and improvement of the entanglement entropy. In particular, the entanglement spectrum contains non-trivial information about the gapless edge states.

Recently, Kim et al. proposed that the thermal Hall conductance can be extracted from the commutator of entanglement Hamiltonians [8]. This conjecture was further generalized to the systems with a global U(1) symmetry by Fan et al. [9]. More specifically, let us consider three jointed regions $A$, $B$ and $C$ of a gapped ground state $|\psi\rangle$ on a two dimensional infinite lattice (all three sectors are large comparing to the correlation length)

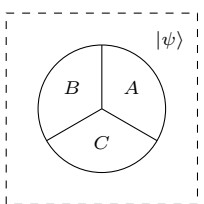

The aforementioned conjectures in Ref. [9] and [8] claim respectively that

$$\sigma_{xy} = \frac{i}{2}\langle\psi|[K_{AB}, Q_{BC}^2]|\psi\rangle, \qquad c_- = \frac{3i}{\pi}\langle\psi|[K_{AB}, K_{BC}]|\psi\rangle. \qquad (1.2)$$

Here $K_{AB} = -\ln \rho_{AB}$ and $K_{BC} = -\ln \rho_{BC}$ are the entanglement Hamiltonians associated with the $AB$ and $BC$ regions; $Q_{BC}$ is the total charge on the $BC$ region. On the l.h.s. of the conjectured formulas, $\sigma_{xy}$ is the electric Hall conductance, and $c_-$ is the chiral central charge[1], i.e. the dimensionless parameter in the thermal Hall conductance $\kappa_{xy} = \frac{\pi}{6}c_- T$. To support the conjectures, the authors of [8–11] provided a few arguments in addition to the numerical tests. However, a rigorous proof is still in search. Part of the difficulty is that the exact form of the entanglement Hamiltonian for a region of irregular shape is hardly tractable.

---

[1]When conformal symmetry is present for the gapless edge, $c_- = c - \bar{c}$ is the difference between the central charges for holomorphic and anti-holomorphic sectors.

Nevertheless, for the ground state of non-interacting fermions, the entanglement Hamiltonian is quadratic and directly related to the spectral projector [12], a matrix that is defined on a single particle Hilbert space rather than an exponentially large many-body Hilbert space. The main result of this paper is a generalized real-space Chern number formula that allows us to compute the commutators of non-linear functions of restricted spectral projectors, including the r.h.s. of (1.2) in their single particle forms as special cases. Our formula is a higher order version of what Kitaev proposed in Ref. [13], known as the "noncommutative Chern number" due to its close connection to the noncommutative geometry [14] and its application in proving the quantization of Hall conductance in disorder systems [15].

The rest of the paper is organized as follows. In Sec. 2, we set up the conventions, review the aforementioned real-space Chern number formula, and show our generalization to higher orders. A combinatorial proof of the generalized formula is presented in Appendix A, and its "smooth" variant is discussed in Appendix B. We then apply the generalized formula to prove the conjectures (1.2) for the non-interacting fermions in Sec. 3 (using a corollary proved in Appendix C). Finally, we conclude with a discussion on the dynamical nature of the commutator formulas with entanglement Hamiltonian in Sec. 4.

# 2 Generalized real-space Chern number formula

## 2.1 Preliminaries

We consider the quadratic Hamiltonian with a spectral gap on a two dimensional infinite lattice

$$H = \sum_{j,k} h_{jk} c_j^\dagger c_k, \quad \{c_j, c_k^\dagger\} = \delta_{jk}, \tag{2.1}$$

where $h = h^\dagger$ is a hermitian matrix.

Assuming a diagonalization $h = UDU^\dagger$, the ground state $|\psi\rangle$ is filled by the normal modes with negative eigenvalues (energies). The spectral projector $P$ is defined accordingly,

$$P := \frac{1}{2}(I - \mathrm{sgn}(h)), \quad \text{with } \mathrm{sgn}(h) := U \mathrm{sgn}(D) U^\dagger. \tag{2.2}$$

The projector $P$ is directly related to the correlation matrix as $\langle\psi|c_j c_k^\dagger|\psi\rangle = \delta_{jk} - P_{jk}$, or equivalently $\langle\psi|c_j^\dagger c_k|\psi\rangle = P_{kj}$. The spectral gap implies a finite correlation length, i.e. $|P_{jk}| < Ce^{-|j-k|/\xi}$ with indices $j,k$ labeling sites on the lattice, and $|j-k|/\xi$ is the distance in the unit of the correlation length $\xi$.

In Ref. [13] (Appendix C. 3), a real-space Chern number formula is proposed as follows

$$\nu(P) = 12\pi i \left[\mathrm{Tr}\left(PAPBPC\right) - \mathrm{Tr}\left(PCPBPA\right)\right], \tag{2.3}$$

where $A, B, C$ are the *spatial* projectors onto the corresponding sectors depicted on the left (i.e. diagonal matrices with diagonal elements 0 or 1 depending on the site indices). All three sectors are large comparing to the correlation length. The value $\nu(P)$ does not change if any

site is reassigned from one sector to another, and therefore is a topological invariant. This definition does not rely on the translational invariance, but in the translational invariant case, it reproduces the familiar TKNN formula expressed in the momentum space [16]

$$\nu(P) = \frac{1}{2\pi i} \int \text{tr}\left(\widetilde{P}\left(\frac{\partial \widetilde{P}}{\partial q_x}\frac{\partial \widetilde{P}}{\partial q_y} - \frac{\partial \widetilde{P}}{\partial q_y}\frac{\partial \widetilde{P}}{\partial q_x}\right)\right) dq_x dq_y, \tag{2.4}$$

where $\widetilde{P}(q_x, q_y)$ is the Fourier transform of $P$ and $\text{tr}(\cdot)$ traces the band indices. As a projector onto the filled bands, $\widetilde{P}(q_x, q_y)$ naturally defines a complex fiber bundle over the torus, on which (2.4) computes the first Chern character.

The Hamiltonian (2.1) is written in the complex fermion basis, which is suitable for systems with charge conservation, such as the integer quantum Hall and the Chern insulators [17]. In these cases, we have the charge operator defined for each local region, e.g., $Q_A = \sum_{j\in A} c_j^\dagger c_j$. The electric Hall conductance and the chiral central charge are proportional to the Chern number as follows

$$2\pi\sigma_{xy} = \nu(P), \qquad c_- = \nu(P) \qquad (P \text{ in complex basis}). \tag{2.5}$$

For systems without a global U(1) symmetry, such as the $(p + ip)$ superconductor [18], a more convenient basis is the Majorana fermions, where a general quadratic Hamiltonian can be written as follows

$$H = \frac{i}{2}\sum_{j,k} m_{jk}\chi_j\chi_k, \qquad \{\chi_j, \chi_l\} = \delta_{jk}. \tag{2.6}$$

Here $m = -m^T$ is a real, skew-symmetric matrix. In the Majorana basis, the spectral projector is given similarly as $P = \frac{1}{2}(I - \text{sgn}(im))$, where $im$ is hermitian and therefore $\text{sgn}(im) = U\,\text{sgn}(D)U^\dagger$ is well defined. Again, we have the relation between the correlation matrix and the projector $\langle\psi|\chi_j\chi_k|\psi\rangle = P_{kj} = \delta_{jk} - P_{jk}$.

As a projector in the Majorana basis, the Chern number of $P$ is defined in the same way as in (2.3). However, the relation to the chiral central charge is different as each Majorana chiral edge mode only carries $1/2$ of the complex fermion one, i.e.

$$c_- = \frac{\nu(P)}{2} \qquad (P \text{ in Majorana basis}). \tag{2.7}$$

Alternatively, one could have directly started with the Majorana basis and regard the charge conserved Hamiltonian (2.1) as a special case. Then the electric Hall conductance, when it applies, comes with an extra $1/2$ factor in its relation to the Chern number, i.e. $2\pi\sigma_{xy} = \nu(P)/2$ with $P$ in the Majorana basis. It is a consequence of the spectrum doubling when we rewrite (2.1) into the form of (2.6).

## 2.2 Generalization

The real-space Chern number formula (2.3) involves the spatial projectors $A$, $B$ and $C$ in a multi-linear way. In this section, we give a generalization that allows repeated appearance of the spatial projectors. The main result is the following formula

$$12\pi i\left[\text{Tr}((PAP)^m PBP(PCP)^n) - \text{Tr}((PCP)^n PBP(PAP)^m)\right] = \frac{6m!n!}{(m+n+1)!}\nu(P) \tag{2.8}$$

where $m, n \in \mathbb{Z}^+$ are positive integers. The original form (2.3) corresponds to the special case when $m = n = 1$.[2] We show a combinatorial proof of (2.8) in Appnedix A.

**Extending to infinite size.** The formula (2.8) is not sensitive to the exterior, since the contribution is localized in the center. Therefore, we may extend $A, B, C$ to the whole infinite lattice (plane) and obtain the following equivalent form using $A + B + C = 1$

$$4\pi i \operatorname{Tr}[(PAP)^m, (PBP)^n] = \frac{2m!n!}{(m+n)!}\nu(P), \qquad (2.9)$$

where $m, n \in \mathbb{Z}^+$ are positive integers. The derivation of the equivalence is contained in the Appendix A (cf. (A.12)). Note it is important that the regions $A$, $B$ are infinite in the above formula, otherwise the commutator is traceless.

**Smooth version.** It is tempting to guess the expressions on the r.h.s. of (2.8) or (2.9) are related to the Euler-Beta function. Indeed, we show in the Appendix B that the following "smooth" version of (2.9)

$$2\pi i \operatorname{Tr}[(PfP)^m, (PgP)^n] = \nu(P) \oint_\Sigma f^m dg^n, \qquad (2.10)$$

also holds. Here $f$ and $g$ are two functions on the two dimensional plane that are smooth over the scale much larger than the correlation length. We further require $f$ and $g$ to have stable asymptotics far away from the center, namely they are functions only of the angular variable at a large enough contour $\Sigma$ (dashed circle in (2.9)), i.e. $f(r, \theta) = f(\theta)$, $g(r, \theta) = g(\theta)$ when outside $\Sigma$. On the l.h.s. of (2.10), $f$ and $g$ are understood as diagonal matrices with elements given by the values of the functions $f$ and $g$.

Heuristically, to mimic (2.9), we let $f_A$ and $g_B$ be two "blurred" indicator functions (i.e. 1 on the corresponding region and 0 otherwise) on $A$ and $B$ respectively, satisfying $f_A + g_B = 1$ along the boundary between $A$ and $B$.

$$(2.11)$$

The non-zero contribution of the integral $\oint_\Sigma f_A^m dg_B^n$ is then concentrated near the blurred boundary (yellow region) and produces the desired answer

$$2\pi i \operatorname{Tr}[(PAP)^m, (PBP)^n] = \int_0^1 (1 - g_B)^m dg_B^n = \frac{m!n!}{(m+n)!}. \qquad (2.12)$$

See Appendix B for more details.

---

[2]The seemingly redundant $P$'s added via $P = P^2$ is to make the combinations $PAP$ and $PCP$ hermitian, so that the powers $m, n$ are analytically continuable.

# 3 Application to entanglement Hamiltonian

The particular generalization (2.8) for the real-space Chern number in the last section was originally motivated by the attempts to prove the two conjectures (1.2) on the relations between the entanglement Hamiltonian and the topological invariants. In this section, we show a proof for the non-interacting gapped fermion system, where the following corollary of (2.8) is vital.

Let $P$ be the spectral projector for a gapped quadratic Hamiltonian of fermions (complex or Majorana) on the two dimensional (infinite) lattice. We have

$$12\pi i \operatorname{Tr}\left(P_{ABC}[P_{AB}^m, P_{BC}^n]\right) = 6\left(\frac{1}{m+n+1} - \frac{m!n!}{(m+n+1)!}\right)\nu(P), \qquad (3.1)$$

for non-negative integers $m, n \in \mathbb{Z}^{\geqslant 0}$. Here the subscripts in the projectors denote the restrictions to the corresponding regions, e.g. $P_{AB} := (A+B)P(A+B)$ where $A$, $B$ are understood as spatial projectors as before. The derivation of the above corollary is presented in Appendix C.

There is an interesting "reflection property" follows immediately from (3.1): if replace $P_{AB}$ by $1 - P_{AB}$, we obtain the same formula with an opposite sign

$$12\pi i \operatorname{Tr}\left(P_{ABC}[(1-P_{AB})^m, P_{BC}^n]\right) = \sum_{m'=0}^{m}(-1)^{m'}\binom{m}{m'} \cdot 12\pi i \operatorname{Tr}\left(P_{ABC}[P_{AB}^{m'}, P_{BC}^n]\right)$$
$$= -6\left(\frac{1}{m+n+1} - \frac{m!n!}{(m+n+1)!}\right)\nu(P). \qquad (3.2)$$

The same reflection property applies when replacing $P_{BC} \to 1 - P_{BC}$.

Furthermore, we also have the following formulas from the derivative of (3.1) at $m = 0$ and/or $n = 0$

$$12\pi i \operatorname{Tr}\left(P_{ABC}[\ln P_{AB}, P_{BC}]\right) = 3\nu(P), \qquad 12\pi i \operatorname{Tr}\left(P_{ABC}[\ln P_{AB}, \ln P_{BC}]\right) = \pi^2 \nu(P). \qquad (3.3)$$

Again, each time when we replace $P_{AB}$ by $(1 - P_{AB})$ or $P_{BC}$ by $(1 - P_{BC})$, the r.h.s flips sign. In the first formula, we have taken $n = 1$ for its application in the next subsection.

## 3.1 Electric Hall conductance

Gapped Hamiltonians with charge conservation and broken time-reversal symmetry can possess chiral edge modes that carry electric current

$$j_{\text{edge}} = \sigma_{xy}\mu, \qquad (3.4)$$

where $\mu$ is the chemical potential that is much smaller than the bulk gap, i.e. $\mu \ll \Delta_{\text{bulk}}$. The dimensionless coefficient $\sigma_{xy}$ is the electric Hall conductance.

Authors of [9] proposed a formula that computes $\sigma_{xy}$ from the ground state wave-function $|\psi\rangle$ as follows,

$$\sigma_{xy} = \frac{i}{2} \langle\psi| [K_{AB}, Q_{BC}^2] |\psi\rangle . \tag{3.5}$$

where $Q_{BC} = \sum_{j \in BC} q_j$ is the charge of the region $BC$ and $K_{AB}$ is the entanglement Hamiltonian for the region $AB$.

The goal of this subsection is to prove (3.4) for the non-interaction gapped fermions. We start with the quadratic Hamiltonian in the form of (2.1), where the local charge is the fermion occupation number, i.e. $Q_{BC} = \sum_{j \in BC} c_j^\dagger c_j$. Moreover, the ground state of such a non-interacting Hamiltonian is Gaussian, i.e. it satisfies Wick's theorem. This key property enables us to write down the entanglement Hamiltonian $K_\Omega$ of the ground state $|\psi\rangle$ for an arbitrary region $\Omega$ [12]

$$K_\Omega = \sum_{jl} k_{\Omega,jl} c_j^\dagger c_l, \qquad k_\Omega = \ln \frac{1 - P_\Omega}{P_\Omega}, \tag{3.6}$$

where $P_\Omega = \Omega P \Omega$ is the spatial restriction of the spectral projector $P$. Spectral projector $P$ is defined in (2.2) and related to the correlation matrix as $P_{jk} = \delta_{jk} - \langle\psi|c_j c_k^\dagger|\psi\rangle = \langle\psi|c_k^\dagger c_j|\psi\rangle$.

Now, we are ready to compute the r.h.s. of (3.5) explicitly via Wick contractions. The result reads

$$\text{r.h.s.} = -i \, \text{Tr} \left( P_{ABC}[k_{AB}, P_{BC}] \right) = -i \, \text{Tr} \left( P_{ABC} \left[ \ln \frac{1 - P_{AB}}{P_{AB}}, P_{BC} \right] \right), \tag{3.7}$$

which can further split into two terms with $\ln(1 - P_{AB})$ and $-\ln P_{AB}$ respectively. Each term can be evaluated as in (3.3), and the sum is $\frac{\nu(P)}{2\pi} = \sigma_{xy}$, which proves (3.5) in non-interacting fermion systems.

## 3.2   Thermal Hall conductance

The chiral edge modes of a gapped ground state with broken time-reversal symmetry can also carry energy, leading to thermal transport

$$I_{\text{edge}} = \frac{\pi}{12} c_- T^2 \tag{3.8}$$

where $I_{\text{edge}}$ is the energy current runs anti-clockwisely along the edge and the temperature is assumed to be much smaller than the bulk energy gap, i.e. $T \ll \Delta_{\text{bulk}}$. As mentioned in the introduction, the dimensionless real number $c_-$ is known as the chiral central charge, which is related to the thermal Hall conductance at low temperature via $\kappa_{xy}(T) = \frac{\pi}{6} c_- T + O(T^2)$.

Authors of [8, 10] proposed the following formula to compute the chiral central charge $c_-$

from the ground state wave-function $|\psi\rangle$

$$c_- = \frac{3i}{\pi} \langle\psi| \, [K_{AB}, K_{BC}] \, |\psi\rangle \qquad (3.9)$$

Since the thermal Hall effect also exists without the charge conservation, it is more convenient to use Majorana basis in this subsection for a unified framework to include both the topological insulators and superconductors. Therefore, we consider the quadratic Hamiltonian in the form of (2.6). The corresponding subregion entanglement Hamiltonian on $\Omega$ is given as follows in the Majorana basis [19]

$$K_\Omega = \frac{i}{2} \sum_{jl} k_{\Omega,jk} \chi_j \chi_l, \qquad ik_\Omega = \ln \frac{1 - P_\Omega}{P_\Omega} \qquad (P \text{ in Majorana basis}) \qquad (3.10)$$

The normalization factor $1/2$ is chosen such that $[-iK(k_{AB}), -iK(k_{BC})] = -iK([k_{AB}, k_{BC}])$. Recall that $\langle\psi|\chi_j\chi_l|\psi\rangle = \delta_{jl} - P_{jl} = P_{lj}$, we have the r.h.s. of (3.9) as follows

$$\text{r.h.s.} = \frac{3i}{2\pi} \text{Tr} \left( P_{ABC} \, [ik_{AB}, ik_{BC}] \right) = \frac{3i}{2\pi} \text{Tr} \left( P_{ABC} \left[ \ln \frac{1 - P_{AB}}{P_{AB}}, \ln \frac{1 - P_{BC}}{P_{BC}} \right] \right), \qquad (3.11)$$

which further splits into four terms involving commutators that have been evaluated in (3.3). The upshot is that each term contributes a $\nu(P)/8$, and they give $\nu(P)/2 = c_-$ in total for $P$ in the Majorana basis.[3]

# 4   Summary and discussion

In this paper, we have generalized the real-space Chern number formula to higher orders, and applied it to establish the relations between entanglement Hamiltonian and Chern number, hence proved the two conjectures (1.2) for non-interacting gapped fermion systems.

A direct question for future is how far we can go in connecting entanglement to topological invariants in the periodic table of free fermion topological states [21–23], e.g., 2d and 3d time-reversal invariant topological insulators [24–26]. Following the strategy of this paper, it seems that suitable generalizations of the real-space Chern number formula to "$\mathbb{Z}_2$" and/or higher dimensions are crucial. Another related open question is how to formulate these conjectures in the low energy effective field theory, such as the Chern Simons theory, where the factorization of Hilbert space is not immediately available.

At last, we would like to comment on the dynamical nature of the commutators that involve an entanglement Hamiltonian. Indeed, as a hermitian operator, the entanglement Hamiltonian $K_\Omega$ of a region $\Omega$ may be used to generate a unitary[4] $U(s) = \exp(-isK_\Omega)$ on $\Omega$. It acts

---

[3]A similar proof has been constructed independently by Nikita Sopenko [20].

[4]Terminology-wise, this evolution $U(s)$ is also known as the (half-sided) modular flow or the modular automorphism. Accordingly, the entanglement Hamiltonian is also called the modular Hamiltonian, due to its appearance in the Tomita-Takesaki modular theory [27–29], whose growing popularity among physicists may be partially attributed to its successful applications in the quantum field theory and the gauge-gravity duality. See [30] for a recent introduction to the subject.

non-trivially on a generic operator $O$ that has support on $\Omega$, via a Heisenberg-like equation

$$O(s) := e^{isK_\Omega}Oe^{-isK_\Omega}, \quad \frac{d}{ds}O(s)\Big|_{s=0} = i[K_\Omega, O], \tag{4.1}$$

where the commutator is interpreted as the rate of change under the evolution $U(s)$. Conjectures (1.2) seem to suggest, as explained in [9, 11] via a Cardy-like formula argument, the above dynamics may be used to distill the universal information from the seemingly non-universal data such as the area-law coefficient $\alpha$ in the von Neumann entropy $S(\rho_A) = \alpha|\partial A| - \gamma + \cdots$. It will be interesting to test this mechanism beyond gapped phases.

It is also worth emphasizing that the dynamics generated by the entanglement Hamiltonian is "intrinsic" – as it is constructed from the entanglement pattern in the wave-function itself rather than external driving sources such as the parent Hamiltonian. This property may be particularly useful when applied to a quantum state that is not obtained from a Hamiltonian but through a quantum simulator [31].

# Acknowledgement

We thank Yiming Chen, Meng Cheng, Daniel Jafferis, Chao-Ming Jian, Charlie Kane, Anton Kapustin, Daniel Parker, Shinsei Ryu, Wanchun Shen, Nikita Sopenko, Ashvin Vishwanath, Huajia Wang and Edward Witten for helpful discussions. We are especially grateful to Alexei Kitaev, whose advice is indispensable for Appendix B. RF is supported by NSF-DMR 2220703 (AV). PZ acknowledges support from the Walter Burke Institute for Theoretical Physics at Caltech. YG is partly supported by the Simons Foundation through the "It from Qubit" program.

# A    Combinatorial proof of the generalized formulas

In this appendix, we present a combinatorial proof of the generalized real-space formula

$$\underbrace{12\pi i\left[\text{Tr}((PAP)^m PBP(PCP)^n) - \text{Tr}((PCP)^n PBP(PAP)^m)\right]}_{=:\alpha_3(m,n)} = \frac{6m!n!}{(m+n+1)!}\nu(P) \tag{A.1}$$

and its equivalent form when $A$, $B$, and $C$ extend to infinity (we will assume $A + B + C = 1$ for the whole section)

$$\underbrace{4\pi i\,\text{Tr}[(PAP)^m, (PBP)^n]}_{=:\alpha_2(m,n)} = \frac{2m!n!}{(m+n)!}\nu(P). \tag{A.2}$$

More specifically, we will derive a set of recurrence relations between $\alpha_3(n, m)$ and $\alpha_2(n, m)$ which will uniquely determine their forms as shown in the above formulas, given the initial condition $\alpha_3(1, 1) = \alpha_2(1, 1) = \nu(P)$.

The derivation is partly inspired by the manipulations in Ref. [13] Appendix C.3.

**A useful lemma.** We begin with a lemma that is useful in deriving the recurrence relation for $\alpha_2(m,n)$. Partition the plane into four different sectors (as shown below), with the corresponding real-space projectors denoted by $\Pi_{1,2,3,4}$ respectively. Then, $\Pi_x = \Pi_1 + \Pi_4$ and $\Pi_y = \Pi_1 + \Pi_2$ are the projectors onto the right and upper half-plane. The lemma is stated as follows

$$\underbrace{2\pi i\,\mathrm{Tr}\left[(P\Pi_x P)^m, (P\Pi_y P)^n\right]}_{=:\rho(m,n)} = \nu(P). \tag{A.3}$$

It reduces to Eq. (128) of Ref. [13] when $m = n = 1$. Therefore, what we need to show is that $\rho(m,n)$ is independent of the arguments $m,n$.

Without loss of generality, we assume $m > 1$, $n \geqslant 1$ and consider the difference

$$\rho(m,n) - \rho(m-1,n) = 2\pi i\,\mathrm{Tr}\left[P(\Pi_2 + \Pi_3)P\big(P(\Pi_1 + \Pi_4)P\big)^{m-1}, \big(P(\Pi_1 + \Pi_2)P\big)^n\right]. \tag{A.4}$$

Now for terms with $\Pi_3$, they contain either (1) two real-space projectors that do not share a boundary; or (2) three or more orthogonal real-space projectors. Both cases have an absolutely convergent trace for individual terms and therefore the trace of the commutator vanishes. Consequently, $\Pi_3$ can be dropped from r.h.s.

$$\text{r.h.s.} = 2\pi i\,\mathrm{Tr}\left[P\Pi_2 P\big(P(\Pi_1 + \Pi_4)P\big)^{m-1}, \big(P(\Pi_1 + \Pi_2)P\big)^n\right]. \tag{A.5}$$

Next, we perform a replacement $\Pi_1 + \Pi_2 = 1 - \Pi_3 - \Pi_4$ and show that the above terms all vanish by repeating the arguments. Similarly, $\rho(m,n) - \rho(m,n-1) = 0$. Q.E.D.

**Recurrence relations.** We will derive three recurrence relations, two of which are for $\alpha_3(m,n)$ and $\alpha_2(m,n)$ separately, and the third one relates $\alpha_3$ and $\alpha_2$.

1. Recurrence relation for $\alpha_3(m,n)$.

   We replace the leftmost/rightmost $A$ in the first/second term of $\alpha_3(m,n)$ with $1 - B - C$ and have
   $$\alpha_3(m,n) = \alpha_3(m-1,n) - \alpha_3(m-1,n+1) \qquad \text{for} \ \ n > 1. \tag{A.6}$$
   We can then start with $\alpha_3(m,1)$ and apply Eq. (A.6) repetitively to deduce

   $$\alpha_3(m,1) = \sum_{j=1}^m \binom{m-1}{j-1}(-1)^{j-1}\alpha_3(1,j) = \sum_{j=1}^m \binom{m-1}{j-1}(-1)^{j-1}\alpha_3(j,1). \tag{A.7}$$

   where in the second step we have used the reflection property, i.e., $\alpha_3(1,j) = \alpha_3(j,1)$, which can be shown by cyclically permuting $ABC$.

2. Recurrence relation for $\alpha_2(n,m)$.

   Starting with the lemma Eq. (A.3), we rewrite it in terms of the real-space projectors $\Pi_{1,2,3,4}$ explicitly

   $$\nu(P) = 2\pi i\,\mathrm{Tr}\left[(P(\Pi_1 + \Pi_4)P)^m, (P(\Pi_1 + \Pi_2)P)^n\right]. \tag{A.8}$$

We can split $(P(\Pi_1 + \Pi_4)P)^m$ into two terms $(P(\Pi_1 + \Pi_4)P)^m - (P\Pi_1 P)^m$ and $(P\Pi_1 P)^m$, each of which has a simple commutator. For the former, since all terms contain $\Pi_4$, we can drop $\Pi_2$ terms in the commutator for the same reason used in the previous subsection. Therefore, we have

$$\mathrm{Tr}\left[(P(\Pi_1 + \Pi_4)P)^m - (P\Pi_1 P)^m, (P(\Pi_1 + \Pi_2)P)^n\right] = \mathrm{Tr}\left[(P(\Pi_1 + \Pi_4)P)^m, (P\Pi_1 P)^n\right]$$
$$= \mathrm{Tr}\left[(P(1 - \Pi_2)P)^m, (P\Pi_1 P)^n\right].$$

(A.9)

From the first to the second line, we have replaced $\Pi_1 + \Pi_4$ with $1 - \Pi_2 - \Pi_3$ and dropped $\Pi_3$. The other term can be manipulated similarly. The upshot is that

$$\nu(P) = 2\pi \mathrm{i}\,\mathrm{Tr}\left[\left(P(1 - \Pi_2)P\right)^m, (P\Pi_1 P)^n\right] + 2\pi \mathrm{i}\,\mathrm{Tr}\left[(P\Pi_1 P)^m, (P(1 - \Pi_4)P)^n\right] \quad (A.10)$$

Now, we are ready to expand the above expression into polynomials of $P\Pi_2 P$ and $P\Pi_4 P$ and obtain the following equation

$$2\nu(P) = -\sum_{j=1}^{m}\binom{m}{j}(-1)^j \alpha_2(j, n) - \sum_{j=1}^{n}\binom{n}{j}(-1)^j \alpha_2(m, j).$$

Taking $m = 1$ and use the reflection $\alpha_2(1, j) = \alpha_2(j, 1)$, we find

$$(1 + (-1)^{m+1})\alpha_2(m, 1) = 2\nu(P) - \sum_{j=1}^{m-1}\binom{m}{j}(-1)^{j+1}\alpha_2(j, 1). \quad (A.11)$$

Note that the left-hand side only involves $\alpha_2(m, 1)$ for odd $m$.

3. The relation between $\alpha_3(m, n)$ and $\alpha_2(m, n)$.

Starting with $\alpha_3(m, n)$, we fully cyclically permute all terms as $(m + n + 1)$-cycles and then use $B = 1 - A - C$ to obtain

$$(m + n + 1)\alpha_3(m, n) = 3\alpha_2(m, n). \quad (A.12)$$

Combining Eq. (A.7) and (A.12), we obtain

$$(1 + (-1)^m)\frac{\alpha_2(m, 1)}{m + 2} = \sum_{j=1}^{m-1}\binom{m-1}{j-1}(-1)^{j-1}\frac{\alpha_2(j, 1)}{j + 2}. \quad (A.13)$$

Note that the left-hand side only involves $\alpha_2(m, 1)$ for even $m$.

**Determining the solution.** First, apply the linear recurrence relation (A.11) and (A.13) for odd and even $m$ respectively, we are able to determine $\alpha_2(m, 1)$ uniquely with initial value $\alpha_2(1, 1) = \nu(P)$. Then, we have $\alpha_3(1, n) = \alpha_3(n, 1) = \frac{n+2}{3}\alpha_2(n, 1)$ from (A.12). Finally, a general $\alpha_3(m, n)$ is achieved via (A.6) and the boundary values $\alpha_3(1, n)$ obtained from the above procedure.

# B  Noncommutative Chern number

The goal of this appendix is to provide a sketch of proof for the "smooth version" of the generalized real-space Chern number formula[5] (cf. Eq. (2.10) in the main text)

$$2\pi i \, \mathrm{Tr}[(PfP)^m, (PgP)^n] = \nu(P) \oint_\Sigma f^m dg^n \, . \tag{B.1}$$

$P$ is the spectral projector for a gapped quadratic Hamiltonian on the two dimensional infinite lattice. The gap condition implies an exponential decaying form of $P$, i.e. $|P_{jk}| < C e^{-|j-k|/\xi}$ with $\xi$ the correlation length. As introduced in the main text, $f$ and $g$ are two functions on the two dimensional infinite lattice that are smooth over scale much larger than the correlation length. We further require $f$ and $g$ to have stable asymptotics far away from the center (for the reason that will be clear momentarily), i.e. $f(r, \theta) = f(\theta)$, $g(r, \theta) = g(\theta)$ when $(r, \theta)$ is outside the contour $\Sigma$. On the l.h.s., $f$ and $g$ are understood as diagonal matrices with elements given by the values of the functions.

To derive (B.1), we consider a version involving three smooth functions and a "bulk integral" on the r.h.s. and then reduce to the above form via Stokes' theorem. More specifically, we show

$$2\pi i \, \mathrm{Tr}\left(Pf_0P\big[(Pf_1P)^m, (Pf_2P)^n\big]\right) = \nu(P) \int f_0 \, df_1^m \wedge df_2^n, \tag{B.2}$$

where $f_{0,1,2}$ are three slow varying functions satisfying $|(L\nabla)^n f_{0,1,2}| \lesssim O(1)$ with $L \gg \xi$, i.e. the typical scale $L$ on which $f_{0,1,2}$ vary is much larger than the correlation length $\xi$.

**Proof sketch of** (B.2). The idea is to use the gradient expansion and show that the error is higher order in $\xi/L$. Let us start with the $m = n = 1$ case,

$$2\pi i \, \mathrm{Tr}\left(Pf_0P[Pf_1P, Pf_2P]\right) = \nu(P) \int f_0 \, df_1 \wedge df_2. \tag{B.3}$$

We divide the two dimensional lattice into patches of size $\Delta x$ by $\Delta y$, with the linear size much larger than the correlation length but much much smaller than $L$, i.e. $\xi \ll \Delta x \sim \Delta y \ll L$. To be concrete, we choose $\Delta x = \Delta y \sim \sqrt{L\xi}$. Accordingly, the trace over all lattice sites can be divided into a sum over the restrictions to individual patches, i.e., $\mathrm{Tr} = \sum_{\mathrm{patch}} \mathrm{Tr}_{\mathrm{patch}}$, where $\mathrm{Tr}_{\mathrm{patch}}(\cdot) := \sum_{j \in \mathrm{patch}} \langle j| \cdot |j\rangle$.

Now, for each patch, we can approximate the restricted trace of the following operators by a global trace with a negligible error

$$2\pi i \, \mathrm{Tr}_{\mathrm{patch}} \left[P\frac{x}{\Delta x}P, P\frac{y}{\Delta y}P\right] = \underbrace{2\pi i \, \mathrm{Tr}\left[P\Pi_x^{\Delta x}P, P\Pi_y^{\Delta y}P\right]}_{=\nu(P)} + O\left(\sqrt{\xi/L}\right) \tag{B.4}$$

---

[5]Many of the discussions in this appendix are inspired by private conversations with Alexei Kitaev, and are also related to the applications of the powerful mathematical tools from noncommutative geometry to the quantum Hall systems [15, 14]. See also Ref. [32] for a more recent overview.

where $\Pi_{x,y}^{\Delta x, \Delta y}$ are smooth versions of the half space projectors

$$\Pi_x^{\Delta x} = \begin{cases} 1 & x > \Delta x \\ x/\Delta x & 0 < x < \Delta x \\ 0 & x < 0 \end{cases} \qquad \Pi_y^{\Delta y} = \begin{cases} 1 & y > \Delta y \\ y/\Delta y & 0 < y < \Delta y \\ 0 & y < 0 \end{cases} \qquad (B.5)$$

which agree with the l.h.s. of (B.4) inside the patch.

To estimate the error, recall that the projector $P$ is local, therefore the discrepancy of the two traces only occurs near the boundary (yellow region) of patch, i.e., in a band of width $\xi$ (the correlation length). Within this band, the difference of the diagonal matrix elements on the two sides is of order $1/(\Delta x \Delta y)$. Hence, the total error is of order $\xi(\Delta x + \Delta y)/(\Delta x \Delta y) \sim O(\sqrt{\xi/L})$. As explained in [13] Appendix C.3, the first term of the r.h.s. of (B.4) is independent of the regulator ($\Delta x$ and $\Delta y$), and equal to $\nu(P)$.

Now, we use (B.4) to evaluate (B.3) patch by patch. Within a patch, the slow varying functions $f_{0,1,2}$ can be linearized with error subleading in $\Delta x/L$. The leading non-zero contribution comes from the constant part of $f_0$ and the first-order derivatives of $f_{1,2}$. We have

$$2\pi i \operatorname{Tr}_{\text{patch}}(P f_0 P [P f_1 P, P f_2 P]) = 2\pi i (f_0 (\partial_x f_1 \partial_y f_2 - \partial_y f_1 \partial_x f_2) \operatorname{Tr}_{\text{patch}}([PxP, PyP]) + O((\xi/L)^{\frac{3}{2}})$$

$$= 2\pi i (f_0 (\partial_x f_1 \partial_y f_2 - \partial_y f_1 \partial_x f_2) \int_{\text{patch}} dx \wedge dy + O((\xi/L)^{\frac{3}{2}})$$

$$= 2\pi i \int_{\text{patch}} f_0 \, df_1 \wedge df_2 + O((\xi/L)^{\frac{3}{2}})$$

$$(B.6)$$

Note that the first term of the r.h.s. is of order $L^{-2} \Delta x \Delta y = \xi/L$, therefore we can ignore the error $O((\xi/L)^{3/2})$. Finally, summing over all the patches yields (B.3).

Generalizing to higher orders (B.2) is straightforward: we divide the trace into the same grid and take only the linear terms in the gradient expansion of $[(P f_1 P)^m, (P f_2 P)^n]$, resulting the r.h.s. of (B.2).

**Reduction to the boundary.** To achieve (B.1) from (B.2), we take $f_0 = 1$ and require $f_1 = f$ and $f_2 = g$ to have stable asymptotics far way from the center, i.e. there exist a circle $\Sigma$ of radius $R$, out of which $f$ and $g$ are only function of angular variable

$$f(r, \theta) = f(\theta), \quad g(r, \theta) = g(\theta) \qquad \text{at } r > R. \qquad (B.7)$$

With this condition, the integrand $df \wedge dg = 0$ vanishes outside of $\Sigma$. Then after applying Stokes' theorem to the integral inside $\Sigma$, we have

$$2\pi i \operatorname{Tr}\left[(PfP)^m, (PgP)^n\right] = \nu(P) \int_{\text{inside } \Sigma} d(f^m dg^n) = \nu(P) \oint_\Sigma f^m dg^n. \qquad (B.8)$$

The $m = n = 1$ case was also discussed in the context of the Girvin-MacDonald-Platzman (GMP) algebra [33–35], where $P$ is the projection to the lowest Landau level.[6]

**Relate smooth to sharp edge.** So far, we have discussed the scenario when functions $f$ and $g$ are smooth enough so that the linear expansion is adequate. In this case, we have a

---
[6]We thank Shinsei Ryu for pointing out this to us.

nice integral formula that computes the desired commutator using the asymptotics of $f$ and $g$. However, in order to apply it to the spatial projectors $A$ and $B$, which are indicator functions with sharp edges, we need extra arguments.

The idea is to design a process that blurs the projectors $A$ and $B$ to $f_A$ and $g_B$ with smoother asymptotics, while keeping the trace of the commutator unchanged, i.e.,

$$2\pi i \operatorname{Tr}\left[(PAP)^m, (PBP)^n\right] = 2\pi i \operatorname{Tr}\left[(Pf_AP)^m, (Pg_BP)^n\right]. \qquad (B.9)$$

The key observation is that when $A + B = 1$ is locally satisfied along the edge, the commutator locally vanishes as $[(PAP)^m, (P(1-A)P)^n] = 0$. Therefore, as long as we preserve the condition $f_A + g_B = 1$ along the edge in the process of deformation, we will have (B.9) satisfied. (One might have concerned that near the center when $A+B = 1$ is not locally satisfied, the argument could have failed. However, since any *finite* change does not contribute to the commutator, the above argument is still valid.)

Now, we are in the position to evaluate the r.h.s. of (B.9) via a simple integral as follows

$$\text{r.h.s.} = \nu(P)\int_0^1 (1-g_B)^m dg_B^n = \frac{m!n!}{(m+n)!}\nu(P) \qquad (B.10)$$

which reproduces (2.9) with a factor of 2 dropped out from both sides.

# C  The commutator of restricted spectral projectors

The generalized real-space Chen number formulas inquire the commutativity between the real-space projectors after the projection. However, for the entanglement Hamiltonian on a region $\Omega$, what appears is the spectral projector restricted to such region, i.e. $P_\Omega = \Omega P \Omega$. In this appendix, we derive the following formula (cf. Eq. (3.1) in the main text) for such restricted spectral projectors,

$$12\pi i \operatorname{Tr}\left(P_{ABC}[P_{AB}^m, P_{BC}^n]\right) = 6\left(\frac{1}{m+n+1} - \frac{m!n!}{(m+n+1)!}\right)\nu(P), \qquad (C.1)$$

as a corollary of the generalized Chern number formulas (2.8) and (2.9). Here $A + B + C$ is a sufficiently large (comparing to the correlation length) but *finite* disk. We emphasize the finiteness of the disk because, in contrast to the generalized Chern number formulas in the main text, the contributions are not localized near the center in the above formula. The key step of the derivation is to rearrange the terms such that the contributions are relocated to the center.

The formula is trivially satisfied for $m = 0$ or $n = 0$. We only need to consider the case for $m, n \in \mathbb{Z}^+$. We start with an expansion of the matrices inside the trace on the l.h.s. of (C.1)

$$\underbrace{P(A+B) \ ... \ P(A+B)}_{m} PB \underbrace{P(B+C) \ ... \ P(B+C)}_{n}$$
$$- \underbrace{P(B+C) \ ... \ P(B+C)}_{n} PB \underbrace{P(A+B) \ ... \ P(A+B)}_{m} \qquad (C.2)$$

1. We separate out the terms in (C.2) that do not contain $C$, i.e.

$$(P(A+B))^m (PB)^{n+1} - (PB)^{n+1}(P(A+B))^m. \tag{C.3}$$

Its trace is zero for the finite disk $A + B + C$ by cyclic permutation. The rest terms are

$$(P(A+B))^m PB\big((P(B+C))^n - (PB)^n\big) - \big((P(B+C))^n - (PB)^n\big)PB(P(A+B))^m. \tag{C.4}$$

2. The next step is to cyclically permute both terms once and add an extra $P$ via $P = P^2$ to the right end of both terms,

$$\begin{aligned}
(P(A+B))^{m-1} PB\big(P(B+C))^n - (PB)^n\big)P(A+\underline{B})P \\
- P(A+\underline{B})\big((P(B+C))^n - (PB)^n\big)PB(P(A+B))^{m-1}P
\end{aligned} \tag{C.5}$$

Note that the $B$ (with a underline and in red color) terms from the first and second line cancel as they are related by cyclic permutations. The additional $P$ was added for later convenience.

3. The rest terms are now supported locally near the center where $A, B, C$ join because all terms will contain at least one copy of $A$, $B$, $C$. Therefore, we can now take the large disk limit and replace $(A+B)$ and $(B+C)$ by $(1-C)$ and $(1-A)$ respectively. We have

$$\begin{aligned}
(P(1-C))^{m-1} PB\big(P(1-A))^n - (PB)^n\big)PAP \\
- PA\big((P(1-A))^n - (PB)^n\big)PB(P(1-C))^{m-1}P.
\end{aligned} \tag{C.6}$$

4. We fully expand the above formula and obtain two series of terms. One is from $(P(1-A))^n$ and is in the form of

$$(PC)^{m'} PB(PA)^{n'+1}P - (PA)^{n'+1}PB(PC)^{m'}P \tag{C.7}$$

with multiplicity $(-1)^{m'+n'}\binom{m-1}{m'}\binom{n}{n'}$. The other is from $(PB)^n$ and is in the form of

$$(PC)^{m'}(PB)^{n+1}PAP - PA(PB)^{n+1}(PC)^{m'}P \tag{C.8}$$

with multiplicity $-(-1)^{m'}\binom{m-1}{m'}$.

5. The trace of both types of terms are computable using (2.8) and (2.9) (with the caution that the powers should be positive while applying (2.8). For the case with power zero, e.g. $m' = 0$ in (C.8), (2.9) is applied instead of (2.8)). The upshot is that

$$\begin{aligned}
12\pi i \, \mathrm{Tr}\left(P_{ABC}[P_{AB}^m, P_{BC}^n]\right) = -6\nu(P)\Bigg[ &\sum_{m'=0}^{m-1}\sum_{n'=0}^{n}(-1)^{m'+n'}\binom{m-1}{m'}\binom{n}{n'}\frac{m'!(n'+1)!}{(m'+n'+2)!} \\
&- \sum_{m'=0}^{m-1}(-1)^{m'}\binom{m-1}{m'}\frac{m'!(n+1)!}{(m'+n+2)!}\Bigg] \\
= -6\nu(P)&\left(\frac{m!n!}{(m+n+1)!} - \frac{1}{m+n+1}\right)
\end{aligned} \tag{C.9}$$

which proves (C.1).

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
