# Peer review of "Generalized Real-space Chern Number Formula and Entanglement Hamiltonian"

_SciPost Physics_

## Round 2 · Referee Report · Anonymous (Referee 2) · 2023-10-18

Strengths

1. Proves two conjectures from other works
2. Is very clearly written

Weaknesses

1. Further discussion on possible applications of the formula would strengthen the manuscript

Report

R. Fan et al. generalize the Real-space Chern number for higher powers of spatial projectors, a result they use to prove two conjectures that relate Hall conductance and central charge to entanglement Hamiltonians.
The mathematical proofs and techniques are clearly presented.
The paper is interesting, and it formalizes results from 2208.11710 and 10.1103/PhysRevLett.128.176402.
Therefore, I would recommend the publication of the work at Scipost Physics once the following questions are addressed:

1. Reference 8, Kim et al., derives the relation between central charge and entanglement Hamiltonian. Could the authors clarify how their proof is more complete than that of Ref. 8?
2. Do the authors consider any other applications that the generalization may allow?
3. Equations 2.8 and 2.9 looks as if the following generalization could be allowed
\begin{equation}
12 \pi i [Tr((PAP)^m(PBP)^k(PCP)^n) - (PCP)^n(PBP)^k(PAP)^m)] = \frac{6 m! n! k!}{m + n + k} \nu(P)
\end{equation}

This equation reduces to 2.8 and 2.9 for $k=1, 0$, respectively. Do the authors think that this generalization could be valid?

Requested changes

1. Could it be that Eq. A.4 misses a minus sign on the right hand side?
2. The manuscript has a typo in page 5, "Appnedix A".
3. I'd like to ask the authors to present the references in a consistent format.

  • validity: high
  • significance: good
  • originality: high
  • clarity: high
  • formatting: excellent
  • grammar: excellent

Author:  Yingfei Gu  on 2023-11-16  [id 4122]

(in reply to Report 2 on 2023-10-18)
Category:
answer to question

We thank the referee for careful reading and valuable comments.

Question: Reference 8, Kim et al., derives the relation between central charge and entanglement Hamiltonian. Could the authors clarify how their proof is more complete than that of Ref. 8?

Reply: We thank the referee for providing us with the opportunity to clarify the relation. In Ref. [8], the authors derived the relation between chiral central charge and modular Hamiltonian with two main non-trivial assumptions. One is the locality of the modular Hamiltonian and the other is an analogy between the modular Hamiltonian and the real physical Hamiltonian. These partially rely on the decomposition of modular Hamiltonians $K_{XY}+K_{YZ}=K_{XYZ}+K_Y$ for sufficiently large subsystem $Y$ when $X$ and $Z$ do not meet directly, and the interpretation of modular Hamiltonians as physical Hamiltonians. However, these assumptions have not been proven rigorously even for systems with finite correlation length $\xi$: $\langle O(\mathbf{x})O(\mathbf{y})\rangle\sim e^{-|\mathbf{x}-\mathbf{y}|/\xi}$. In our work, we provide a rigorously proof for the relation for non-interacting fermions with finite correlation length without any assumption for the modular Hamiltonian or other symmetries, which complements the original derivation in Ref. [8].

Question: Do the authors consider any other applications that the generalization may allow?

Reply: We thank the referee for raising the question. There is a similar question in Report 3 and we would like to answer them together. The calculation of non-linear response functions or higher-moment fluctuations necessarily involve operators with high powers. Our formula can have potential applications in this case. Our formula is potentially useful to establish other properties of the free-fermion entanglement, e.g. the recently proposed conjecture on the relation between reflected entropy and total edge central charge (which is not shown rigorously yet even for free fermion systems).

Question: Equations 2.8 and 2.9 looks as if the following generalization could be allowed \begin{equation} 12\pi i \left[ {\rm Tr} ((PAP)^m (PBP)^k (PCP)^n-(PCP)^n (PBP)^k (PAP)^m ) \right] = \frac{6m!n!k!}{(m+n+k)!} \nu(P) \end{equation} This equation reduces to 2.8 and 2.9 for $k=1,0$, respectively. Do the authors think that this generalization could be valid?

Reply: We thank the referee for this valuable question. Although this formula is correct for $k=0,1$, it turns out to be wrong for general $k$. Nevertheless, the smooth version is correct: \begin{equation} 12\pi i \left[ {\rm Tr} ((Pf_0P)^m (Pf_1P)^k (Pf_2P)^n-(Pf_2P)^n (Pf_1P)^k (Pf_0P)^m ) \right] = \frac{6m!n!k!}{(m+n+k)!} \nu(P), \end{equation} where $f_{0,1,2}$ are three slow varying functions satisfying $|(L\nabla)^n f_{0,1,2}| \lesssim O(1)$ with $L\gg \xi$, i.e. the typical scale $L$ on which $f_{0,1,2}$ vary is much larger than the correlation length $\xi$. This can be proved by firstly repeating the derivation in Appendix B, which gives \begin{equation}\notag l.h.s.= 6 \nu(P) \int f_0^m ~df_1^k \wedge df_2^n= \frac{6m!n!k!}{(m+n+k)!} \nu(P)=r.h.s. \end{equation} The second equality is derived under the condition $f_0+f_1+f_2=1$. If we attempt to relate results of smooth edges to sharp edges as in Appendix B, we have to show that any finite change near the center does not contribute to $ {\rm Tr}((Pf_0P)^m (Pf_1P)^k (Pf_2P)^n-(Pf_2P)^n (Pf_1P)^k (Pf_0P)^m) $. However, this holds true only for $k=0,1$. We have also verified our statements above through numerics.

Requested changes: 1. Could it be that Eq. A.4 misses a minus sign on the right hand side? 2. The manuscript has a typo in page 5, "Appnedix A".

Reply: We would like to express our gratitude to the referee for bringing the typos to our attention. We have fixed them in the revised manuscript.

Requested changes: I'd like to ask the authors to present the references in a consistent format.

Reply: We have revised our reference list following the suggestion of the referee.

---

## Round 2 · Referee Report · Anonymous (Referee 3) · 2023-10-26

Strengths

- Mathematically elegant proofs of two recent conjectures.
- Provides a new higher-order version of the noncommutative Chern number.

Weaknesses

- The paper could have had a better motivation and more highlight on the importance of proving the conjectures (1.2).
- Potential applications of the generalized Kitaev's Chern number formula beyond the proofs of (1.2) not mentioned.

Report

The authors provide a unified proof of two recent conjectures that link the Hall conductivity and the chiral central charge to the modular (or entanglement) Hamiltonian in free fermionic systems, based on a higher-order version of the Kitaev formula for the noncommutative Chern number that they also introduce and prove.

The proofs of the formulas relating Hall conductivity and chiral central charge to the entanglement Hamiltonian are of significant interest and are mathematically elegant and correct.

I recommend the manuscript for publication in SciPost Physics.

Minor comment:
The generalized Chern number formula in the paper is only used as a lemma in order to prove (1.2). It would be valuable to investigate further properties of the higher Chern number formula. In particular, could the authors comment on potential applications of this formula to the numerical computation of Chern number for finite-size free fermion systems? Could the generalization (2.8) have any numerical advantage over the usual Chern marker formula m=n=1 ?

Typos:
Page 7: "The goal of this subsection is to prove (3.4) for the non-interaction gapped fermions." -> "The goal of this subsection is to prove (3.5) for the non-interacting gapped fermions."

Requested changes

Fix typos.

  • validity: high
  • significance: high
  • originality: high
  • clarity: high
  • formatting: good
  • grammar: good

Author:  Yingfei Gu  on 2023-11-16  [id 4121]

(in reply to Report 3 on 2023-10-26)
Category:
answer to question

We thank the referee for careful reading and inspiring comments.

Question: The generalized Chern number formula in the paper is only used as a lemma in order to prove (1.2). It would be valuable to investigate further properties of the higher Chern number formula. In particular, could the authors comment on potential applications of this formula to the numerical computation of Chern number for finite-size free fermion systems? Could the generalization (2.8) have any numerical advantage over the usual Chern marker formula $m=n=1$?

Reply: We thank the referee for raising the question. Since there is a similar question in Report 2, we copy our answer here. The calculation of non-linear response functions or higher-moment fluctuations necessarily involve operators with high powers. Our formula can have potential applications in this case. Our formula is potentially useful to establish other properties of the free-fermion entanglement, e.g. the recently proposed conjecture on the relation between reflected entropy and total edge central charge (which is not shown rigorously yet even for free fermion systems).

Question: Page 7: "The goal of this subsection is to prove (3.4) for the non-interaction gapped fermions." -$>$ "The goal of this subsection is to prove (3.5) for the non-interacting gapped fermions."

Reply: We thank the referee for pointing out our typo. We have fixed it in the revised manuscript.

Summary of changes made

  1. We have fixed the typos mentioned in both reports.

  2. We have improved the reference list as suggested by Report 2.

---

## Editorial Decision

resubmitted